# Mistargeted retinal axons induce a synaptically independent subcircuit in the visual thalamus of albino mice

Sean McCracken[1,2,3,4], Liam McCoy[1,2,4], Ziyi Hu[4], Julie A Hodges[1,2,4], Katia Valkova[1,2,4], Philip R Williams[1,2,3], Josh L Morgan[1,2,4]*

[1]John F Hardesty, MD Department of Ophthalmology and Visual Sciences, Washington University School of Medicine, St. Louis, United States; [2]Department of Neuroscience, Washington University School of Medicine, St. Louis, United States; [3]Hope Center for Neurological Disorders, Washington University School of Medicine, St. Louis, United States; [4]Biomedical Engineering, Washington University School of Medicine, St. Louis, United States

*For correspondence: jlmorgan@wustl.edu

Competing interest: The authors declare that no competing interests exist.

## eLife Assessment

This study provides **important** observations about the role of Hebbian synapse rewiring (which predicts that correlated activity between neurons begets stronger synapses) on brain connectivity development by examining a naturally emerging case where Hebbian predictions can be tested because neurons with differing activity undergo development under otherwise similar conditions (albino mouse lateral geniculate nucleus [LGN], where retinal ganglion cells [RGCs] from the contra-lateral retina form inappropriate projections alongside a majority of ipsilateral RGCs). The evidence supporting the conclusions is **compelling**, with combined confocal imaging and serial electron microscopy (EM) reconstructions showing complete synaptic isolation of aberrantly projecting RGCs onto LGN thalamocortical projection neurons, and mixed connectivity onto LGN local interneurons. The morphological descriptions of connectivity presented here will be of interest to researchers studying synaptic connectivity and plasticity during development.

**Abstract** In albino mice and EphB1 knockout mice, mistargeted retinal ganglion cell axons form dense islands of axon terminals in the dorsal lateral geniculate nuclei (dLGN). The formation of these islands of retinal input depends on developmental patterns of spontaneous retinal activity. We reconstructed the microcircuitry of the activity-dependent islands and found that the boundaries of the island represent a remarkably strong segregation within retinogeniculate connectivity. We conclude that when sets of retinal input are established in the wrong part of the dLGN, the developing circuitry responds by forming a synaptically isolated subcircuit within the otherwise fully connected network. The fact that there is a developmental starting condition that can induce a synaptically segregated microcircuit has important implications for our understanding of the organization of visual circuits and our understanding of the implementation of activity-dependent development.

## Introduction

Activity-dependent synaptic remodeling is a process in which patterns of neuron depolarization and synaptic transmission determine whether a given synapse will be maintained, enhanced, or elimi-nated (*Katz and Shatz, 1996*; *Lichtman and Colman, 2000*; *Redfern, 1970*; *Lichtman, 1977*; *Crepel*

*et al., 1976*). This process is important for both the formation of circuits during development and the creation of long-term memory. However, it has proven difficult to perform experiments in which specific patterns of neural activity can be mapped onto features of the synaptic organization of circuits. To understand how experience shapes the cellular organization of circuits, we look to a model system in which stereotyped patterns of activity generate a stereotyped functional architecture.

In amniotes (birds, reptiles, mammals), the dorsal lateral geniculate nucleus (dLGN) provides the most direct link between the retina and visual cortex. The dLGN's functional architecture is similar to that of their immediate upstream circuit, the inner plexiform layer of the retina. Both neuropils consist of multiple topographic maps of visual space stacked together as functionally distinct layers (*Reese, 1988*). Unlike the architecture of the retina, the functional architecture of the dLGN depends heavily on an extended period of activity-dependent synaptic remodeling (*Chen and Regehr, 2000*; *Stellwagen and Shatz, 2002*; *Lee et al., 2002*; *Godement et al., 1984*; *Hahm et al., 1991*). Blocking the spontaneous waves of activity disrupts the refinement of dLGN retinotopy (*Grubb et al., 2003*; *Pfeiffenberger et al., 2006*) and the normal segregation of left and right eye inputs (*Feller, 2009*).

The innervation of the dLGN by the retina starts with molecular gradients guiding axons to roughly the correct location (*Sitko et al., 2018*) where they form synapses with many more thalamocortical cells (TCs) than they will innervate in the adult (*Chen and Regehr, 2000*). The retina then generates spontaneous waves of activity in which retinal ganglion cells (RGCs) with similar positions and receptive field properties fire together (*Meister et al., 1991*). The temporal correlation of this RGC firing is thought to help refine the innervation of the dLGN according to Hebbian rules (*Butts et al., 2007*; *Lee et al., 2002*; *Morris and Hebb, 1949*). By selecting a set of RGC inputs that fire at the same time, TCs can ensure that their inputs are driven by a single region of space and have similar functional properties. The result is a refined map of visual space where nearby TC share functional properties and, often, RGC inputs (*Morgan et al., 2016*).

What would happen if a small set of RGC axons took a wrong turn early in development and found themselves surrounded by RGC axons with very different activity patterns? The Hebbian model would predict that these inputs might not be integrated into the surrounding network. Rather, they might capture a set of TCs that is exclusive to the mistargeted RGCs. The result would be a synaptically segregated retinogeniculate subcircuit, an organization not normally observed in mouse dLGN (*Morgan et al., 2016*).

Examination of the RGC boutons distributions in mice in which the initial targeting of RGCs to the dLGN is disrupted reveals structures that are consistent with the predicted segregated microcircuit. In albino mice (B6(Cg)-*Tyr*$^{c-2J}$/J) (*Rebsam et al., 2012*) and EphB1-knockout mice (*Rebsam et al., 2009*), sets of RGC axons that would normally project to the ipsilateral dLGN instead project to the contralateral dLGN. Some of these mistargeted axons form islands of intense bouton labeling within the dLGN. Blocking developmental retinal waves prevents the formation of these islands (*Rebsam et al., 2009*; *Rebsam et al., 2012*). These islands of RGC boutons are, therefore, strong candidates for being the segregated microcircuits predicted by a Hebbian model of the results of axon mistargeting.

What was unknown, based on these previous studies, was the microcircuitry underlying the islands of dense RGC bouton labeling. Here, we use correlated light to electron microscopy to test whether the islands of boutons in the albino mouse dLGN form synapses or are simply a tangle of axon terminals that could not find a target. We find that RGC axons in the island form the complex retinogeniculate synaptic glomeruli typical of normal dLGN. We then test the Hebbian prediction that TCs should be innervated by either the island or non-island RGC boutons, but not by both. We found that the island does, in fact, reflect a segregated microcircuit where mistargeted axons capture an exclusive set of TCs. Finally, we examined the extent to which the retinogeniculate circuit segregation extends to other cell types in the albino dLGN. Our results demonstrate not only that the activity-dependent RGC island constitutes a segregated microcircuit, but that the segregation has a significant impact on the cellular organization of the dLGN.

## Results

### Albino island is surrounded by an RGC bouton exclusion zone

We first replicated previous studies of the distribution of RGC boutons in the albino mice. We labeled the RGCs of 12 albino mice with the anterograde neuronal tracer choleratoxin-B (CtB) and examined

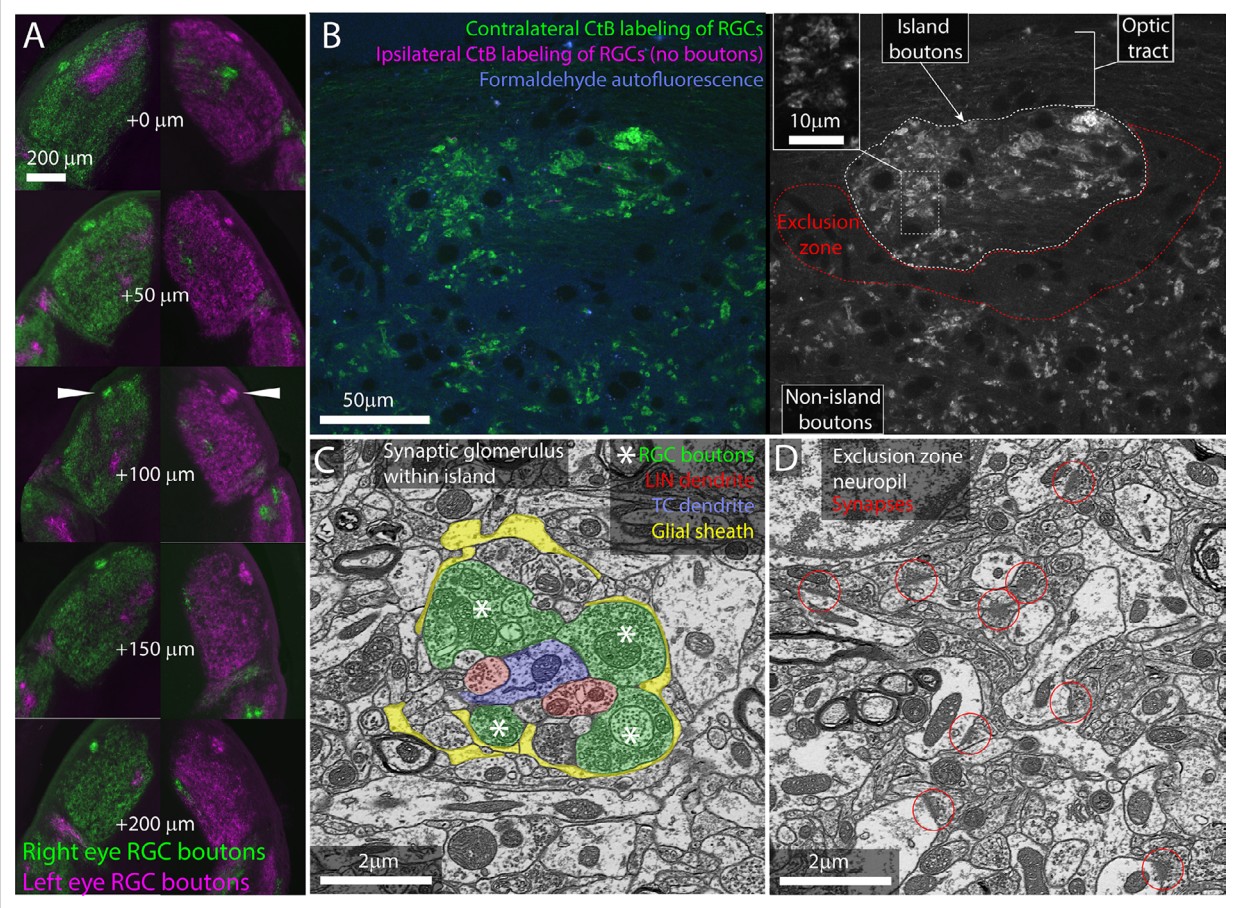

**Figure 1.** The island of retinal ganglion cell (RGC) boutons represents a segregation and concentration of a subset of otherwise normal retinogeniculate connections. (**A**) Serial vibratome sections through the left and right dorsal lateral geniculate nuclei (dLGN) of an albino mouse. From top to bottom, section progress rostral to caudal. Choleratoxin-B (CtB) labeling of RGCs from the left (magenta) and right (green) eyes. White arrowheads indicate island. (**B**) Single confocal section through CtB-labeled albino island. Left shows background fluorescence (blue), contralateral RGC boutons (green), and an absence of ipsilateral axons (magenta). Right shows CtB-labeled RGC boutons from contralateral eye only. Inset shows enlarged view of RGC boutons. (**C**) Electron micrograph of retinogeniculate glomerulus in the albino island. RGC boutons are indicated by asterisks. (**D**) Electron micrograph of non-glomerular neuropil of the exclusion zone. Red circles indicate non-glomerular synapses. Both the synaptic glomerulus in the island and the feedback synapses in the exclusion zone appear ultrastructurally normal.

The online version of this article includes the following figure supplement(s) for figure 1:

**Figure supplement 1.** Synaptophysin labeling of dorsal lateral geniculate nuclei (dLGN) synapses shows synaptic neuropil is present in the retinal ganglion cell (RGC) bouton exclusion zone.

the dLGNs with widefield epifluorescence imaging of serial vibratome sections. CtB labeling is particularly bright within synaptic boutons, and injecting each eye with a different color of fluorescently bound CtB (Alexa 488, 568, 647) is a standard method of revealing retinogeniculate synaptic connectivity.

Consistent with previous studies, we found that the dLGNs of all 12 mice examined showed bright, distinct islands of contralaterally projecting RGC boutons (*Figure 1*).

We next examined the cellular organization of the albino island in the same samples using confocal microscopy. Confocal imaging of the CTB-labeled albino dLGNs showed that CtB labeling of RGC boutons within the island resembled labeling of RGC boutons outside of the island with the exception that the density of labeling was higher within the island (*Figure 1B*). Both inside and outside of the island, we observed the fluorescent profile of bright dots surrounding a dark spot. This profile is consistent with synaptic glomeruli in which labeled RGC boutons clustered around unlabeled TC dendrites (*Figure 1B and C*).

Confocal imaging also made it clear that the island was surrounded by an RGC-bouton-free region, which we term the exclusion zone. Examination of this exclusion zone with autofluorescence, reflected

light, and immunolabeling for synaptophysin (*Figure 1B*, *Figure 1—figure supplement 1*) demonstrates that the exclusion zone was composed of a mix of neuropil, cell nuclei, myelinated fibers, and an occasional blood vessel. We did not detect an anatomical barrier that would preclude RGC boutons from forming in this region.

We next asked whether the optically labeled island boutons formed normal synaptic connections. We used correlated light and electron microscopy (*Friedrichsen et al., 2022*) to examine the ultrastructure of one albino island. We found that RGC boutons in the island participated in normal retinogeniculate synaptic glomeruli (*Figure 1C*). RGC boutons were identified by their light mitochondria, large size, and large synaptic vesicles (*Hamos et al., 1987*). Thalamocortical dendrites were distinguished from local inhibitory neuron dendrites by the presence of spines and the absence of synaptic outputs. RGC boutons were found in clusters surrounding TC dendrites. The RGC boutons form normal synaptic triads innervating local inhibitory neuron boutons that then form inhibitory synapses onto TCs. As in normal dLGN, the cluster of RGC and local inhibitory neuron boutons are encapsulated by a glial sheath (*Saavedra and Vaccarezza, 1968*).

Consistent with the CtB labeling, electron microscopy (EM) of the exclusion zone revealed large regions of dLGN neuropil with no RGC boutons (*Figure 1D*). Aside from the lack of RGC glomeruli, the exclusion zone appeared healthy. Normal mouse dLGN neuropil can be divided into RGC-bouton-associated glomerular neuropil (*Figure 1C*, colored) and non-glomerular neuropil. The non-glomerular neuropil consists primarily of the distal dendrites of TCs and their inputs from the visual cortex and thalamic reticular nucleus (*Sherman and Guillery, 1996*). The synaptic neuropil of the exclusion zone was composed of non-glomerular synapses and their associated neurites (*Figure 1D*).

Our examination of RGC boutons in the dLGN of the albino mouse, therefore, rules out the possibility that the islands are dead-end neuromas buried in the optic tract. Rather, the bright islands of CtB labeling observed in the albino dLGN represent a high concentration of ultrastructurally normal RGC boutons embedded in healthy dLGN neuropil.

## An exclusive set of TCs are captured by the island RGC boutons

Does the spatial segregation of RGC boutons reflect a segregation in the functional connectivity of the albino dLGN? TCs are the sole output neuron of the dLGN, and their receptive field properties are largely the product of which RGCs innervate them (*Usrey et al., 1998*). The pattern of RGC innervation of TCs, therefore, defines the functional architecture of the dLGN. In previous EM reconstructions of mouse dLGN circuitry, we found that nearby TCs tend to be innervated by the same RGC axons (*Morgan et al., 2016*). By contrast, If the exclusion zone around the RGC island represents a segregation of retinogeniculate circuitry, we would expect to find a hard boundary in connectivity. TCs near the exclusion zone would be innervated by either the island or non-island RGC boutons, not both.

To test this hypothesis, we collected a serial section EM volume to reconstruct the arbors and synaptic connectivity of TCs near the exclusion zone. The EM volume was obtained from an optically characterized albino dLGN and consisted of 2631 40-nm-thick coronal sections (*Figure 2A and B*). From each section, we acquired a 220 μm × 220 μm image mosaic that encompassed most of the albino island, the exclusion zone, and the surrounding non-island dLGN neuropil (105 μm of rostro-caudal z-depth). Images were acquired with a 20 nm pixel size. For analysis, the voxel size was treated as 26 nm × 20 nm × 40 nm to compensate for tissue compression during cutting. The large pixel size (usually 4 nm for circuit reconstruction) allowed for rapid acquisition of a large vEM volume while still allowing for the tracing of large features such as TC dendrites and RGC boutons (*Figure 2C*).

We traced 15 TCs whose nuclei were distributed across the RGC island exclusion zone (*Figure 2B*). TC dendrites were traced until they left the volume or until they had clearly transitioned into a distal dendrite morphology. All reconstructions should be considered partial. The transition from proximal to distal dendrite morphology was identified as the point where dendrites become thin and unbranched and form spines that are innervated by small cortical feedback boutons as opposed to RGC boutons. After tracing the TCs, we labeled the RGC boutons innervating the TCs.

The RGC inputs identified in the EM reconstruction of TCs covered approximately a third of the island visible in the confocal imaging of the same tissue (*Figure 2D*). 3D rotation of the reconstructed boutons revealed a clear exclusion zone between the reconstructed island boutons and the surrounding boutons (*Figure 2E*). The exceptions were RGC boutons associated with TC17. This TC

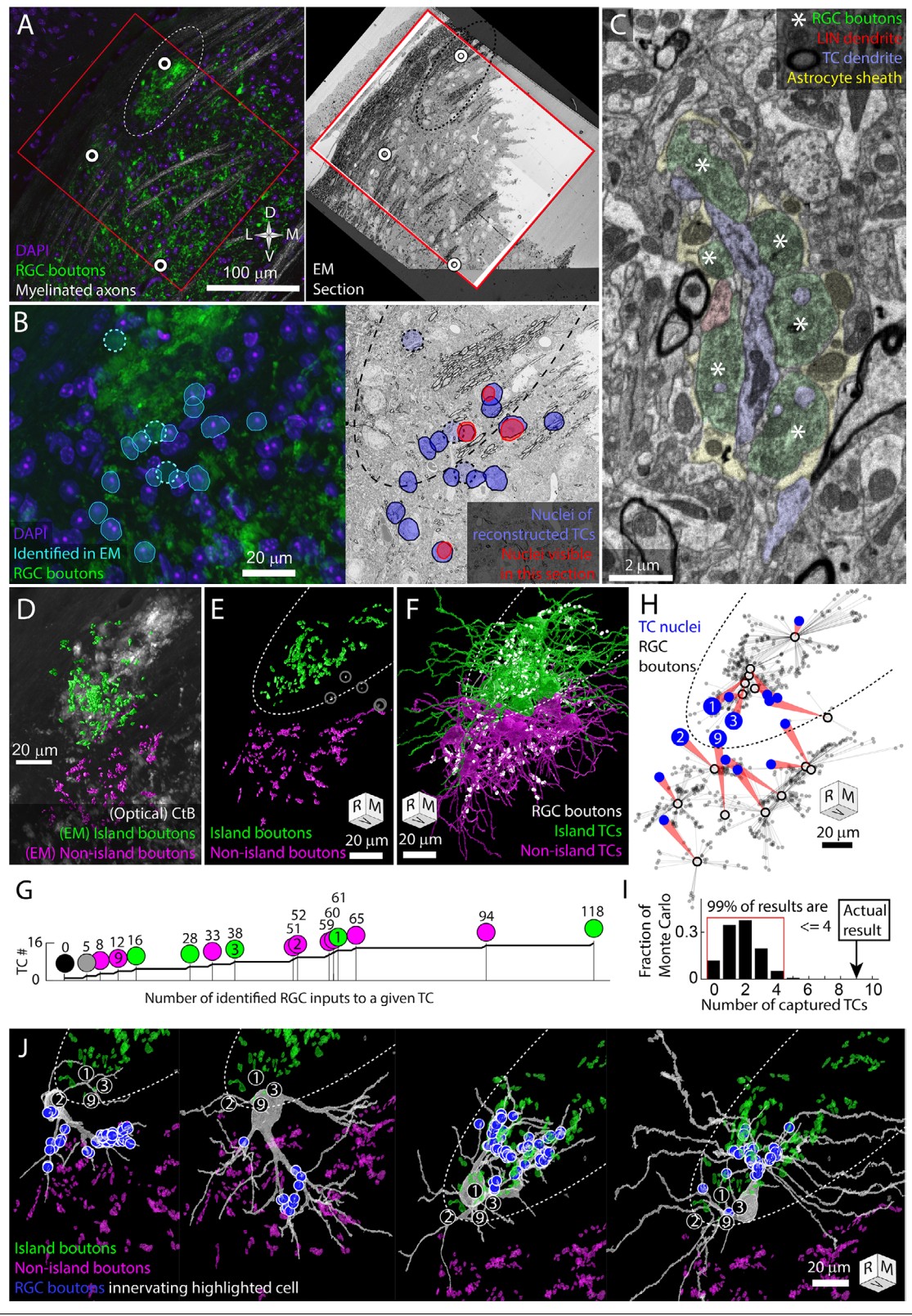

**Figure 2.** The retinal ganglion cell (RGC) island represents a segregation of retinogeniculate synaptic connectivity. Thalamocortical cells are either innervated by island RGC boutons or non-island RGC boutons. Dotted oval indicates the boundary of the RGC island in multiple panels. Four cells (TC1, TC2, TC3, TC9) are identified in (**B, G, H, I**). Tissue orientation is indicated by compass rose or rotation cube (rostral/medial/ventral). (**A**) Matching of optical images and EM section overview. Red rectangles indicate the boundaries of the EM volume used to trace thalamocortical cells

*Figure 2 continued on next page*

*Figure 2 continued*

(TCs). White circles indicate blood vessels matched between optical images and EM images. (**B**) Closer view of light and EM images showing nuclei of EM-reconstructed TCs. In the optical image (left), DAPI-labeled nuclei (blue) of reconstructed TCs are outlined (cyan) relative to RGC boutons (green). The position of reconstructed TCs that are not visible in the confocal image stack is indicated by circles. The nuclei labels are mapped onto the EM section (right). The nuclei of reconstructed TCs visible in the EM section are highlighted in red. (**C**) Example of electron micrograph with 20 nm pixel size in which TC dendrite and associated glomerulus are annotated. RGC boutons are indicated by asterisks. (**D**) EM reconstructed island (green) and non-island (magenta) RGC boutons that were found innervating reconstructed TCs. EM boutons are overlayed on the corresponding optical image of choleratoxin-B (CtB)-labeled RGC boutons. (**E**) Rotation of EM-labeled RGC boutons to show the clearest separation between island (green) and non-island boutons (magenta). Boutons innervating the one TC without a clear island or non-island identity are highlighted in white. (**F**) Partial reconstructions of TCs indicated in (**B**). TCs are color coded by whether the receive their input from the island or non-island RGC boutons. Positions of RGC inputs are indicated by white dots. (**G**) Cumulative curve of the number of RGC boutons innervating reconstructed TCs. Circles indicate island/non-island identity. Numbers above the circles are the number of RGC boutons innervating the TC. Number within the circle indicates TC IDs used in other panels. (**H**) Plot of the relationship between TC body (blue circle) and the location of the RGC inputs that innervate it (gray dots). Average synapse position is indicated by a black circle. (**I**) Results of Monte Carlo simulation predicting how many TCs would be innervated only by island or only by non-island RGC boutons if TC dendrites are independent of one another. Red bracket encloses 99% of results. (**J**) EM reconstructions of the four example TCs highlighted in previous panels. Circled numbers indicate the relative position of the four example cells. The location of RGC inputs is indicated by blue circles. Island and non-island RGC boutons innervating other TCs shown in green and magenta. TCs with soma in the exclusion zone have asymmetric dendritic arbors that reflect their exclusive connectivity to either the island or non-island RGC boutons.

The online version of this article includes the following video and figure supplement(s) for figure 2:

**Figure supplement 1.** Thalamocortical cells (TCs) surrounding the exclusion zone are captured by either the island or non-island retinal ganglion cell (RGC) boutons.

**Figure supplement 2.** Proximal dendrites of thalamocortical cells (TCs) are usually thick and innervated by retinal ganglion cell (RGC) boutons.

**Figure 2—video 1.** 3D rotation of thalamocortical cells (TCs) shown in *Figure 2—figure supplement 1*.

https://elifesciences.org/articles/100990/figures#fig2video1

was innervated by only five RGC boutons, three of which were in the middle of the exclusion zone (*Figure 2E* gray circles).

We found that 14 TCs were innervated either exclusively by the island (six TCs) or exclusively by RGC boutons outside of the island (eight TCs, *Figure 2F–H*). There were no mixed non-island/island TCs. There was one TC (mentioned above) that received inputs in the exclusion zone and one TC on which no RGC inputs could be found. The wide range of the number of RGC boutons innervating TCs (0–118, *Figure 2G*) may reflect the fact that, in the dorsal mouse dLGN, synaptic inputs from the superior colliculus can take the place of RGCs as a primary driving synapse (*Bickford et al., 2015*).

We next compared our observed results for the rate of complete capture of a TC to what we would expect if there was no preference for being innervated by only island or only non-island RGC boutons. We designed a simple Monte Carlo simulation based on the nine TCs whose cell bodies were most clearly in the exclusion zone and that were innervated by the island or non-island RGC boutons. We posited that, in the absence of preference, each dendrite emerging from the cell body would have an independent probability of connecting to either island or non-island RGC boutons. We therefore counted the number of dendrites that emerged from each TC body that eventually connected to an RGC input (2,2,3,4,4,4,5,5,7). We then ran 100,000 trials in which each primary dendrite was assigned island or non-island connectivity with equal probability. For each iteration, we counted the number of TCs that were innervated either only by island boutons or only by non-island boutons (captured). We found that 99% of the Monte Carlo iterations produced four (44%) or fewer captured TCs (*Figure 2I*). The nine captured TCs (100%), therefore, represent a substantial deviation from an unbiased assignment of dendrites to island or non-island RGCs.

The absence of TCs that were innervated both by the island and non-island RGC boutons means that the island represents a segregation of retinogeniculate synaptic connectivity. This exclusivity is what we had predicted based on the model in which activity-dependent synaptic remodeling would prevent strong functional dissimilarity in the RGC innervation pattern of TCs. We were surprised, though, at the extent of dendritic arbor asymmetry we observed associated with this connectivity pattern. Asymmetric TC arbors, particularly in the dorsal shell of the dLGN, are not surprising (*Krahe et al., 2011*). What was noteworthy was that the asymmetry was so clearly oriented away from the exclusion zone.

Few TC dendrites crossed into the exclusion zone and those that did assumed a distal dendrite morphology (*Figure 2H and J*, *Figure 2—figure supplements 1 and 2*, *Figure 2—video 1*). While

most of these distal morphology neurites were left untraced, more complete tracings of two TCs confirmed that these dendrites maintained their non-RGC-receiving character when they passed out of the exclusion zone (TC3, TC4, *Figure 2J*, *Figure 2—figure supplements 1 and 2*). The presence of cortical feedback synapses (small round boutons innervating small spines, *Guillery, 1971*) on the dendrites that span the exclusion zone means that the cortical feedback onto the TCs does not obey the strict island/non-island segregation exhibited by the RGC innervation.

## Local inhibitory neurons follow RGC segregation

In mouse, RGC inputs to TCs are almost always coupled with connections to local inhibitory neurons (LINs) (*Morgan et al., 2016*). These inhibitory neurons form three types of neurites: (1) input/output shaft dendrites that span hundreds of micrometers and multiple subregions of the dLGN, (2) input/output targeted dendrites that extend about 20 µm from the shaft dendrites and closely follow nearby RGC axons, and (3) output-only axons that are relatively small (*Morgan and Lichtman, 2020*). The short distances between the output synapses of the targeted neurites and multiple RGC inputs mean the targeted neurites can provide a synaptic drive that reflects the functional properties of the local dLGN neuropil (*Morgan and Lichtman, 2020*). Output synapses on the shaft dendrites of LINs, in contrast, are expected to be driven more by the global integration of inputs across the large LIN dendritic arbor. Based on these properties, we hypothesized that the shaft neurites of local inhibitory neurons would be indifferent to the boundaries of the RGC island while individually targeted neurites would be associated with either island or non-island RGC boutons. We tested this hypothesis with both light and EM reconstructions of local inhibitory neurons.

In GAD67-GFP transgenic mice, dLGN local inhibitory neurons are brightly labeled with GFP (*Seabrook et al., 2013*; *Charalambakis et al., 2019*). We bred these mice with the albino *Tyr-/-* mice to generate GAD67-GFP, *Tyr-/-* transgenic mice (*Figure 3A*). Airyscan confocal images of the island revealed LIN dendrites that were closely associated with the CtB labeling of RGC boutons (*Figure 3B*). This tight association with RGC boutons is characteristic of the targeted dendrites of LINS. In contrast, the shaft dendrites of LINs crossed the RGC bouton exclusion zone without obvious deviation (*Figure 3A and C*). This pattern was consistent across five mice examined.

We used our vEM volume to determine if the impression of shaft dendrite indifference to island boundaries and targeted dendrite avoidance of island boundaries was reflected in synaptic connectivity. We first selected one LIN for reconstruction based on the presence of its nucleus in the exclusion zone. This LIN extended dendrites into both the island and non-island neuropil and formed synapses with RGCs and TCs in both regions (*Figure 3D*). We then selected three LIN dendrites for tracing based on their synaptic connectivity with identified TCs and RGC boutons. Two of these tracings revealed long targeted neurites that formed many connections to TCs and RGCs specifically within either the island or non-island neuropil (*Figure 3E*).

In the third cell, we found seven dendritic processes extending from a small region near the interface of the island and the exclusion zone (*Figure 3F*). Four of the dendrites crossed the exclusion zone, formed no synapses with RGCs or TCs in this region, and then formed synapses with RGCs and TCs when they entered the non-island neuropil. One of these dendrites exhibited the thin diameter, tortuous path, and dense glomerular innervation of a LIN-targeted dendrite. This dendrite was interesting in that it received 4 island boutons near its base and then received 42 synapses from non-island RGC boutons.

The LIN results are consistent with our prediction that shaft dendrites would be indifferent to island/non-island boundaries while individual targeted dendrites would target either the island or non-island RGC boutons. However, the restriction of the targeted dendrites to one or the other RGC field does not appear to be an absolute rule. Rather the scale of targeted dendrite exploration and the size of the exclusion zone are likely to reduce the chances that a targeted dendrite would find partners on both in the island and outside of the island. This matching between the exploration of targeted LIN dendrites and the segregation of retinogeniculate connectivity means that targeted LIN dendrites will have an RGC input profile (island/non-island) that matches the TCs they innervate.

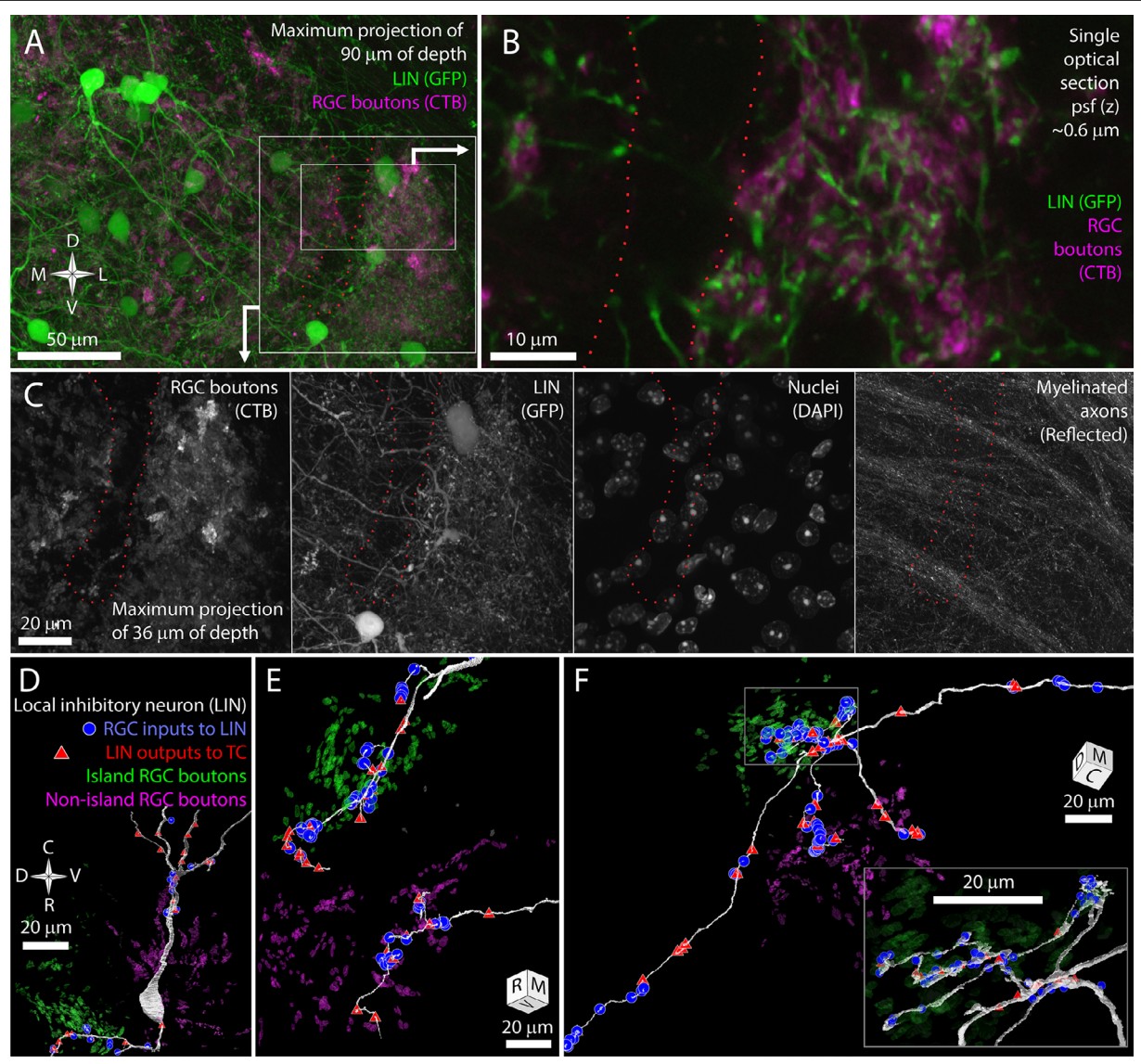

**Figure 3.** The long shaft dendrites of LINs cross the exclusion zone while the individual targeted dendrites of LINs have distinct island or non-island synaptic identities. (**A**) Reference image for (**B, C**) showing LINs (green) and choleratoxin-B (CtB) labeling of retinal ganglion cell (RGC) terminals (magenta). (**B**) Close association between targeted neurites of LINs (green) and RGC terminals (magenta). (**C**) Four channels of confocal stack showing RGC bouton exclusion zone relative to LINs, cell nuclei, and reflective fibers. (**D**) EM reconstruction of LIN relative to island (green) and non-island (magenta) RGC boutons. Input synapses from RGCs (blue circles) and output synapses (red triangles) are labeled. (**E**) Two LIN dendrites labeled as in (**D**). (**F**) LIN dendrite labeled as in (**D**). Inset shows a magnified view of branch points. The functional identities of LINs are, therefore, likely to be locally, but not globally, specific to island or non-island boutons.

## Discussion

The key conclusion of this study is that the previously reported islands of mistargeted RGC terminals represent a segregation of retinogeniculate circuitry into distinct pathways. The extent of this segregation was surprising. In a previous EM reconstruction of a wildtype mouse dLGN, nearby TCs shared input from the same RGC axons even when the RGC axons or TCs were of morphologically distinct types (*Morgan et al., 2016*). Based on these results, we argued that the parallel pathways of different types of RGCs did not have synaptically clean boundaries in the mouse dLGN, but instead were frequently intermixed. In contrast, the island/non-island projections appear truly parallel (non-intersecting). The fact that there is a developmental starting condition that can induce a synaptically

segregated microcircuit has important implications for our understanding of the organization of visual circuits and for our understanding of the implementation of activity-dependent development.

## Network organization of mouse dLGN

In cats and primates, the receptive field properties of TCs are usually derived from a strong innervation from a few functionally similar RGCs (*Usrey et al., 1998*; *Mastronarde, 1992*; *Hamos et al., 1987*). This connectivity allows for a spatially continuous sampling of visual space with little loss in acuity (*Alonso et al., 2006*; *Martinez et al., 2014*). Studies of retinogeniculate connectivity in mouse suggested a messier view of dLGN processing. Mouse TCs can be innervated by dozens of RGCs (*Morgan et al., 2016*; *Hammer et al., 2015*; *Rompani et al., 2017*), and different RGC types innervate the same TC (*Liang et al., 2018*; *Marshel et al., 2012*; *Rompani et al., 2017*). Part of the explanation of the discrepancy is that the mouse dLGN contains both high- and low-convergence TCs (*Morgan et al., 2016*; *Rompani et al., 2017*). Further, many of the RGC inputs to a given mouse TC are much weaker than the relatively few RGCs that dominate the TC firing (*Litvina et al., 2017*). Within the circuitry of the mouse dLGN, therefore, there are TCs whose connectivity is consistent with the gated relays of retinal activity described in other species.

However, the most striking connectivity rule that came out of our mouse EM reconstruction was that we could find no distinction between TCs (morphology, convergence, ultrastructure) that precluded two TCs from being innervated by the same RGC axon (*Morgan et al., 2016*). We wondered if the seeming lack of synaptic specificity was because mouse evolution valued signal detection over functional specificity, the mixing of channels was a statistical selection of related channels, or synaptic specificity was limited by the size of the tissue. A whole mouse dLGN can fit inside one of the six functionally distinct sublamina of a macaque dLGN (*Casagrande et al., 2007*). The odd case of the albino island eliminates the possibility that the mixing of retinogeniculate connections observed in the normal mouse reflects an upper limit of the structural specificity of the system. The mechanisms exist in the mouse dLGN to generate completely segregated subcircuits. The normal network organization, therefore, likely reflects an adaptive balance between receptive field refinement, space, and the benefits of integrating signals from multiple RGCs.

## Evidence for activity-dependent remodeling as the driving force behind islands

How much of the circuit segregation of the albino dLGN islands can we attribute to activity-dependent remodeling? There are activity-independent mechanisms that could be contributing to the island phenotype. Molecular recognition helps keep RGC axons from similar regions of the dLGN bundled together as they travel through the optic tract (*Sitko et al., 2018*; *Bruce et al., 2017*). Homotypic membrane recognition molecules are also required for RGC boutons to cluster together to form synaptic glomeruli (*Monavarfeshani et al., 2018*). A set of mistargeted RGC axons of the same type might, therefore, be expected to stay bundled together in the absence of activity-dependent cues.

On the other hand, there is extensive evidence showing that developmental patterns of retinal activity are critical for shaping the architecture of retinogeniculate connections (*Liang and Chen, 2020* for review). The waves of activity that propagate through the retina prior to eye opening are particularly important for eye-specific segregation (*Feller, 2009*) and topographic refinement (*Grubb et al., 2003*; *Pfeiffenberger et al., 2006*). Blocking the propagation of these waves also prevents the formation of RGC bouton islands in albino mice and EphB1 KO mice (*Rebsam et al., 2012*; *Rebsam et al., 2009*). Thus, the segregation of island and non-island RGC boutons depends on the same activity-dependent synaptic remodeling process that is responsible for refining the receptive fields of TCs in the normal mouse dLGN.

The connectivity of the RGC islands reported here also supports the conclusion that activity-dependent circuit remodeling generates the islands. First, the island is not a neuroma of tangled RGC axons, and the exclusion zone is not a region of pathological dLGN. Rather, the island and exclusion zone represent a segregation or retinogeniculate connectivity within an otherwise normal dLGN neuropil. While activity-independent signals might drive dissimilar RGC boutons apart, it does not explain why a TC could not be innervated by both island and non-island RGC boutons. The simplest explanation for the clustering of RGC boutons, the binary connectivity of TCs, and the systematic

asymmetry of the TC dendritic arbors is that the island is the result of activity-dependent Hebbian synaptic remodeling.

## A model of activity-dependent synaptic remodeling

The process of activity-dependent synaptic remodeling begins with RGC axons having already positioned themselves in roughly the correct region of the dLGN (*Godement et al., 1984*). However, each TC receives input from many more RGCs than will eventually drive it (*Chen and Regehr, 2000*) and its receptive field properties are likewise less refined than they will be in the adult (*Tavazoie and Reid, 2000*; *Tschetter et al., 2018*). The dLGN then undergoes a period of activity-dependent remodeling in which spatio/temporally structured retinal activity is thought to provide the information TCs need in order to select which retinal inputs to maintain and which to eliminate (*Hong and Chen, 2011*).

Unlike many of the model systems where synapse elimination has been well characterized (*Redfern, 1970*; *Crepel et al., 1976*; *Lichtman, 1977*), the synaptic remodeling of TCs is not all or none. Functionally similar RGCs can maintain and elaborate their connections on the same TC. The capacity of functionally similar synapses from different axons to reinforce one another may result from the fact that the activity-dependent synaptic enhancement of retinogeniculate synapses acts on the time scale of bursts of activity (~1 s) instead of individual spikes (*Butts et al., 2007*). A mature RGC can, therefore, share a TC target with a slightly more ventral RGC, which shares a target with an even more ventral RGC and so on in every direction. The result of dLGN remodeling is a refined but also synaptically continuous remapping of visual space.

In albino mice, where sets of RGCs make large and correlated errors in the initial targeting of retinogeniculate projections, synaptic stabilization is no longer equal in every direction. A group targeting error produces sets of axons whose firing patterns are strongly correlated within the group but that are different from the firing patterns of surrounding RGC axons. At the start of synaptogenesis, the dLGN neuropil at the boundary between the mistargeted and normally targeted RGC axons would be composed of TC dendrites whose RGC inputs are systematically less correlated than dendrites oriented away from the boundary. Mistargeted RGCs would be more likely to stabilize synapses in the center of the field innervated by the mistargeted RGCs. This process would also be self-reinforcing. As boundary synapses are weakened in favor of centrally located synapses, centrally located TC targets become even more attractive than boundary targets. The mechanism driving mistargeted retinogeniculate connections to collapse into a completely segregated island would not be different from the mechanisms of normal activity-dependent development. The difference is simply that in the normal dLGN, synaptic stabilization forces are much closer to being equal in all directions.

## Why is there an exclusion zone?

Hebbian synaptic remodeling predicts mistargeted axons will capture a set of unique TC targets, but it does not predict the spatial segregation of synapses we observe in the form of the exclusion zone. The cell bodies of island and non-island innervated TCs are close enough together that their RGC-bouton-receiving dendrites should overlap. If the only remodeling rule implemented was to prune the weaker population of synapses on TCs innervated by both island and non-island RGCs, then we would expect to find the spared RGC boutons filling the boundary region between the mistargeted and normally targeted RGC axons. The exclusion zone requires an additional level of circuit remodeling.

An explanation for the exclusion zone is suggested by the asymmetry of the TC dendritic arbors relative to the exclusion zone. The asymmetry would seem to require that developing TCs direct dendritic resources toward favored inputs. TC dendrites normally respond to RGC inputs by extending spines around input RGC boutons (*Wilson et al., 1984*). If TCs are deprived of retinal inputs during development, they undergo a brief period of exuberant dendritic growth followed by dendritic pruning (*El-Danaf et al., 2015*). In adult dLGN, loss of RGC inputs leads to an eventual shrinkage of dendritic arbors (*Bhandari et al., 2022*). If the trophic interaction between RGC boutons and TC dendrites can operate with dendritic specificity, then the exclusion zone could represent a region where dendrites innervated by functionally disparate RGC inputs were pruned in favor of dendrites in regions where RGC inputs were correlated.

## Materials and methods

### Animals

Albino mice (B6(Cg)-*Tyr^c-2j*/J, JAX:000058) and GAD67-GFP (CB6-Tg(Gad1-EGFP)G42Zjh/J, JAX:007677) mice were obtained from Jackson labs. All mice were between 2 and 3 months old. Optical examination of dLGNs included both male and female mice. The mouse on which vEM was performed was male. All procedures in this study were approved by the Institutional Animal Care and Use Committee of Washington University School of Medicine (protocol # 23-0116) and complied with the National Institutes of Health Guide for the Care and Use of Laboratory Animals.

To obtain tissue, mice were transcardially perfused with either 4% paraformaldehyde in 0.1 M phosphate-buffered saline (PBS) (optical imaging only) or 2% paraformaldehyde and 1% glutaraldehyde in 0.1 M PBS. Brains were removed and post fixed for 30 minutes to an hour. Brains were then cut to 50–200-μm-thick coronal sections using a Compresstome (Precisionary Instruments) vibratome.

### Optical labeling

RGC axons were labeled by intraocular injection of CtB bound to either Alexa-488, Alexa-555, or Alexa-647. Cell nuclei were labeled by incubating vibratome sections in DAPI for 20 minutes or by mounting in Fluoromount with DAPI (Thermo Fisher Scientific). Synapses were immunolabeled with anti-synaptophysin (1:160, S5768). Vibratome slices were partially cleared by soaking in 2,2′-thiodiethanol (TDE) or by repeated applications and removal of Vectashield (Vector Laboratories).

### Optical imaging

Brain slices were first imaged using a widefield epifluorescence microscope (Leica). Slices were then imaged using an FV-1000 confocal microscope (Olympus). Scans typically included two channels of CtB fluorescence acquisition, a DAPI channel, a reflected light channel (no laser filtering), and an autofluorescence channel (excitation wavelength for aldehyde and without corresponding label). Images of GAD67-GFP-labeled LINs were acquired with a Zeiss LSM 800 (Zeiss) confocal microscope in Airyscan mode. Images were acquired with low laser, maximum scan speed, and averaging between 3 and 8 scans. High-resolution images were acquired with a 60 ×1.4 NA oil objective. For presentation, the brightness, contrast, and gamma of each channel have been independently adjusted. The goal of this adjustment is to make both the tissue background signal and cellular structures visible without saturating pixels.

### Electron microscopy

After optical imaging of the full vibratome brain slice, tissue was removed from the slide and refixed with 2% paraformaldehyde and 2% glutaraldehyde. The dLGNs were then isolated from the rest of the brain slice and washed with 0.1 M cacodylate buffer. Tissue then treated with osmium tetroxide -> ferrocyanide -> thiocarbohydrazide -> osmium tetroxide -> uranyl acetate -> lead aspartate (*Hua et al., 2015*; *Morgan et al., 2016*). 40-nm-thick ultrathin sections were collected using an automatic tape-collecting microtome (RMC ATUM, Leica UC7 ultramicrotome; *Schalek et al., 2012*). Sections were collected onto carbon-coated Kapton (gift from Jeff Lichtman lab) and then mounted onto silicon wafers.

Wafers were mapped on a Zeiss Merlin scanning electron microscope using WaferMapper (*Hayworth et al., 2014*). Mapping includes acquiring a low-resolution image of each tissue section. The resulting image volume was compared with optical images of the tissue section to target further imaging. Before high-resolution images were acquired, sections were plasma etched with an ibss plasma asher (ibss Group, Inc) to enhance contrast and surface stability (*Morgan et al., 2016*). High-resolution (2 nm pixel) images were acquired from targeted regions of the island and exclusion zone. The vEM volume used for most of the circuit reconstruction was acquired at 20 nm pixel size. Both types of images were acquired with the in-lens secondary electron detector at 1 keV landing voltage and 1–3 nA. The 3D image volume was aligned by ScalableMinds.

### vEM rendering and analysis

Circuit reconstruction was performed via manual tracing in VAST (*Berger et al., 2018*). Segmentations were imported from VAST to CellNav (https://github.com/MorganLabShare/CellNav, copy archived at

*MorganLabShare, 2025a*) for analysis and visualization. 3D renderings and analysis can be regenerated by downloading CellNav version AlbinoIsland and the AlbinoIsland CellNav cell library (https://sites.wustl.edu/morganlab/data-share/). On executing 'runCellNav', select the AlbinoIsland CellNav directory. All cells and synapses included in this study are available to view. The rendering of the island and non-island boutons can be generated by Menu -> RunScripts -> Publication -> AlbinoIsland -> jm_KxS_showGreenMagentaBoutons.m. The images used in *Figure 2—figure supplement 1* can then be generated by running Menu -> RunScripts -> Publication -> AlbinoIsland -> jm_KxS_show-CellsAndRGC_snapShots.m. The plots in *Figure 2* can be generated by running Menu -> RunScripts -> Publication -> AlbinoIsland -> PlotCBtoSynVector. Synapses associated with particular cells can be rendered by selecting a cell or cell type from the Select Cells panel and then clicking 'Cell' or 'Type' in the Groups panel. The cell group can then be defined as the pre- or post-synaptic cell in the 'Synapses' tab. Synapses can be filtered by pre- or post-synaptic cell type use the 'preClass' or 'postClass' drop-down menus.

### Monte Carlo analysis

We performed a Monte Carlo analysis to gauge the extent to which the connectivity of the reconstructed circuit deviates from a circuit in which there is no selection against a TC being innervated by both island and non-island RGC boutons. We selected nine reconstructed TCs whose cell bodies were determined to be in the exclusion zone. For these TCs, we counted how many primary dendrites were innervated by RGC boutons or whose downstream dendrites were innervated by RGC boutons. We then ran a model 100,000 times in which each primary dendrite had an equal probability to be connected to the island or non-island RGC boutons. We then counted how many TCs in each trial were innervated by only the island boutons or only the non-island boutons. We reported the value (four TCs) for which 99% of the Monte Carlo results produced equal or fewer captured TCs.

### Statistical design

In this project, we have tested the hypothesis that the mistargeted RGC axons in albino mice form a synaptically segregated retinogeniculate circuit. This hypothesis was qualitative. Prior to performing this study, we did not have a reasonable basis for formulating a quantitative hypothesis regarding how much synaptic segregation would constitute a segregated microcircuit. We, therefore, did not design a null hypothesis significance test to reject the null hypothesis that the circuit was not segregated.

We did compare our observed circuit reconstruction to a model of a circuit without segregation and found that our EM reconstruction demonstrated a substantial deviation from the no-segregation prediction. This model is based on nine cells from the same piece of tissue. There is no control for variation between individuals. Nonetheless, the test demonstrates that it is possible for a mouse dLGN to produce a highly segregated retinogeniculate subcircuit. We also report that the optically identifiable correlate to the structures we describe in our EM reconstruction is present in 12 of 12 albino mice examined and match the pattern reported in the previous literature (*Rebsam et al., 2012*).

## Acknowledgements

We thank the Washington University Center for Cellular Imaging and staff where the EM was performed and to ScalableMinds that aligned the EM dataset. We also thank Richard Schaleck and the Jeff Lichtman lab for providing carbon-coated Kapton collection tape. This work was supported by an unrestricted grant to the Department of Ophthalmology and Visual Sciences from Research to Prevent Blindness, a Research to Prevent Blindness Career Development Award (JLM and PRW), and the NIH (EY029313 to JLM, R01EY032908 to PRW, T32 EY013360 to SM).

## Additional information

### Funding

| Funder | Grant reference number | Author |
|---|---|---|
| National Institutes of Health | EY029313 | Liam McCoy<br>Julie A Hodges<br>Katia Valkova<br>Josh L Morgan |
| Research to Prevent Blindness | Career development award | Philip R Williams<br>Josh L Morgan |
| Research to Prevent Blindness | Unrestricted grant to the Department of Ophthalmology and Visual Sciences | Sean McCracken<br>Liam McCoy<br>Julie A Hodges<br>Katia Valkova<br>Philip R Williams<br>Josh L Morgan |
| National Institutes of Health | R01EY032908 | Philip R Williams |
| National Institutes of Health | T32 EY013360 | Sean McCracken |

The funders had no role in study design, data collection and interpretation, or the decision to submit the work for publication.

### Author contributions
Sean McCracken, Data curation, Writing – review and editing; Liam McCoy, Data curation, Investigation; Ziyi Hu, Julie A Hodges, Katia Valkova, Investigation; Philip R Williams, Supervision, Funding acquisition, Writing – review and editing; Josh L Morgan, Conceptualization, Resources, Data curation, Software, Formal analysis, Supervision, Funding acquisition, Investigation, Visualization, Methodology, Writing – original draft, Project administration, Writing – review and editing

### Author ORCIDs
Josh L Morgan (iD) https://orcid.org/0000-0002-2657-2416

### Ethics
All procedures in this study were approved by the Institutional Animal Care and Use Committee of Washington University School of Medicine (Protocol # 23-0116) and complied with the National Institutes of Health Guide for the Care and Use of Laboratory Animals.

Reviewer #1 (Public review): https://doi.org/10.7554/eLife.100990.3.sa1
Reviewer #2 (Public review): https://doi.org/10.7554/eLife.100990.3.sa2
Author response https://doi.org/10.7554/eLife.100990.3.sa3

## Additional files

### Supplementary files
MDAR checklist

### Data availability
All code used in this publication is available from GitHub at https://github.com/MorganLabShare/albinoIsland2025 (copy archived at *MorganLabShare and Mahsa-jmorganLab, 2025b*). Segmentations of EM data are available on Dryad at DOI: https://doi.org/10.5061/dryad.j3tx95xs2.

The following dataset was generated:

| Author(s) | Year | Dataset title | Dataset URL | Database and Identifier |
|---|---|---|---|---|
| McCracken S, McCoy L, Hu Z, Hodges J, Valkova K, Williams PR, Morgan J | 2025 | Albino Island CelNav_KxS | https://doi.org/10.5061/dryad.j3tx95xs2 | Dryad Digital Repository, 10.5061/dryad.j3tx95xs2 |

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
