## [Editor Report · eLife Assessment]

This study provides **important** observations about the role of Hebbian synapse rewiring (which predicts that correlated activity between neurons begets stronger synapses) on brain connectivity development by examining a naturally emerging case where Hebbian predictions can be tested because neurons with differing activity undergo development under otherwise similar conditions (albino mouse lateral geniculate nucleus [LGN], where retinal ganglion cells [RGCs] from the contralateral retina form inappropriate projections alongside a majority of ipsilateral RGCs). The evidence supporting the conclusions is **compelling**, with combined confocal imaging and serial electron microscopy (EM) reconstructions showing complete synaptic isolation of aberrantly projecting RGCs onto LGN thalamocortical projection neurons, and mixed connectivity onto LGN local interneurons. The morphological descriptions of connectivity presented here will be of interest to researchers studying synaptic connectivity and plasticity during development.

---

## [Referee Report · Reviewer #1 (Public review)]

Summary:

The authors examined whether aberrantly-projecting retinal ganglion cell in albino mice innervate a separate population of thalamocortical neurons, as would be predicted for Hebbian learning rules. The authors find support for this hypothesis in CLEM reconstructions of retinal ganglion cell axons and thalamocortical neurons. In a second line of investigation, the authors ask the same question about retinal ganglion cell innervation of local inhibitory neurons of the mouse LGN. The authors conclude that these connections are less specific.

Strengths:

Good use of CLEM to test a circuit-level hypothesis

Interesting difference between TC and LIN neurons found

Weaknesses:

The authors have addressed all concerns in the last round to my satisfaction.

---

## [Referee Report · Reviewer #2 (Public review)]

In this article, the authors examined the organization of misplaced retinal inputs in the visual thalamus of albino mice at electron-microscopic (EM) resolution to determine whether these synaptic inputs are segregated from the rest of the retinogeniculate circuitry.

The study's major strengths include its high resolution, achieved through serial EM and confocal microscopy, which enabled the identification of all synaptic inputs onto neurons in the dorsolateral geniculate nucleus (dLGN).

The experiments are very precise and demanding thus, only the synaptic inputs of a few neurons were fully reconstructed in one animal.

Despite this, the authors clearly demonstrate the synaptic segregation of misrouted retinal axons onto dLGN neurons, separate from the rest of the retinogeniculate circuitry.

This finding is impactful because retinal inputs typically do not segregate within the mouse dLGN, and it was previously thought that this was due to the nucleus's small size, which might prevent proper segregation. The study shows that in cases where axons are misrouted and exhibit a different activity pattern than surrounding retinal inputs, segregation of inputs can indeed occur. This suggests that the normal system has the capacity to segregate inputs, despite the limited volume of the mouse dLGN.

---

## [Author Response]

The following is the authors’ response to the original reviews.

**Reviewer #1 (Public review):**
Summary:The authors examined whether aberrantly projecting retinal ganglion cells in albino mice innervate a separate population of thalamocortical neurons, as would be predicted for Hebbian learning rules. The authors find support for this hypothesis in correlated light and electron microscopy (CLEM) reconstructions of retinal ganglion cell axons and thalamocortical neurons. In a second line of investigation, the authors ask the same question about retinal ganglion cell innervation of local inhibitory interneurons of the mouse LGN. The authors conclude that these connections are less specific.Strengths:The authors make good use of CLEM to test a circuit-level hypothesis, and they find an interesting difference in RGC synaptic innervation patterns for thalamocortical neurons vs. local interneurons.Weaknesses:The conclusions about the local interneuron innervation are a little more difficult to interpret. One would expect to only capture a small part of the local interneuron dendritic field, as compared to the smaller thalamocortical neurons, right? Doesn't that imply that finding some evidence of promiscuous connectivity means that other dendrites that were not observed probably connect to many different RGCs?

We will try to clarify this point

**Reviewer #2 (Public review):**
In this article, the authors examined the organization of misplaced retinal inputs in the visual thalamus of albino mice at electron-microscopic (EM) resolution to determine whether these synaptic inputs are segregated from the rest of the retinogeniculate circuitry.The study's major strengths include its high resolution, achieved through serial EM and confocal microscopy, which enabled the identification of all synaptic inputs onto neurons in the dorsolateral geniculate nucleus (dLGN).The experiments are very precise and demanding; thus, only the synaptic inputs of a few neurons were fully reconstructed in one animal. A few figures could be improved in their presentation.Despite this, the authors clearly demonstrate the synaptic segregation of misrouted retinal axons onto dLGN neurons, separate from the rest of the retinogeniculate circuitry.This finding is impactful because retinal inputs typically do not segregate within the mouse dLGN, and it was previously thought that this was due to the nucleus's small size, which might prevent proper segregation. The study shows that in cases where axons are misrouted and exhibit a different activity pattern than surrounding retinal inputs, segregation of inputs can indeed occur. This suggests that the normal system has the capacity to segregate inputs, despite the limited volume of the mouse dLGN.
**Recommendations for the authors:**

**Reviewer #1 (Recommendations for the authors):**
(1) Please include page numbers and line numbers in future submissions.

Done

(2) I am red-green colorblind, and I had a lot of trouble seeing the red channels when they were mixed with green. I recommend using magenta when possible.

Thanks for the heads up. We have switched to green and magenta where possible. In the tinted EM where switching colors did not seem helpful, we added an asterisk to RGC boutons so that red and green would not be the only identifiers.

(3) It would help if the figure captions also stated the conclusions that can be drawn from the figures. I recommend stating the main conclusion in the first sentence of the caption, rather than stating only what we are viewing. Similarly, the last sentence of the caption can help summarize what has been seen.

We have included summary sentences at the beginning and end of figure legends.

(4) In the text when discussing Figure 2J, do the authors mean to cite Supplementary Figure 2?

Yes, thanks.

(5) I don't think TC was ever defined (or I didn't find it).

Corrected

(6) In the subsection "An exclusive set..." cite Liang et al. as more evidence of non-specific innervation.

We cite Liang et al in the discussion, but I don’t see a good place to cite it in the referenced results section. Please elaborate if we are missing something.

(7) Supplementary Figure 3 is never cited.

We have added the citation to Figure 3.

(8) I found myself unsure of what to conclude after the results on LIN. A few more sentences of interpretation and restating what was found would help.

We have added additional clarification in the Results:

“The LIN results are consistent with our prediction that shaft dendrites would be indifferent to island/non-island boundaries while individual targeted dendrites would target either the island or non-island RGC boutons. However, the restriction of the targeted dendrites to one or the other RGC field does not appear to be an absolute rule. Rather the scale of targeted dendrite exploration and the size of the exclusion zone is likely to reduce the chances that a targeted dendrite would find partners on both in the island and outside of the island. This matching between the exploration of targeted LIN dendrites and the segregation of retinogeniculate connectivity means that targeted LIN dendrites will have an RGC input profile (island/non-island) that matches the TCs they innervate.”

**Reviewer #2 (Recommendations for the authors):**
(1) The abbreviation TC is used in the text without a definition.

Corrected

(2) The features that allow for labeling the different dendrites/cells (TC and LIN) in Serial EM data (Figure 1) are necessary. While the explanation is provided for RGC boutons, the labeling for thalamic cells is not discussed.

We added the sentence:

“Thalamocortical dendrites were distinguished from local inhibitory neuron dendrites by the presence of spines and the absence of synaptic outputs.”

(3) Image 2C (EM) appears blurry or pixelated. Enhancing its resolution could improve clarity.

Image 2C is a demonstration of how much we felt we could sacrifice image quality and still reconstruct TC arbors and RGC inputs.

(4) The gray circles that show the innervation of TC17 in Figure 2E are barely visible, especially on-screen without high magnification. A more contrasting color and wider lines would enhance visibility. It would also be helpful to indicate TC17 in Figure 2H and 2G, as this cell is special and highlighted in the main text.

We have made the requested changes

(5) A TC with no RGC input is mentioned. Have you identified other synaptic inputs, potentially related to SC or the cortex?

Both TC17 (a few exclusion zone RGC inputs) and TC5 (no RGC inputs) were innervated by some large, dark mitochondria boutons that could be SC inputs. However, we did not perform enough reconstruction of the axons to confidently describe their non-RGC input profile. I have previously observed occasional TCs in the same region of the dLGN where RGC inputs are almost entirely replaced by SC inputs, so finding two such cells was not surprising.

(6) Two fully reconstructed TCs are mentioned. Please specify their exact number in the text, as citing Figure 2J or Supplementary Figure 1 alone is not sufficient for identification.

Clarified as “(TC3, TC4, Figure 2J, Supplementary Figure 2,3).”

(7) A correlation between the position of the dendrites and the location of RGC inputs would provide additional insights. This is somewhat reminiscent of the dendrite orientation of Layer IV spiny stellate neurons in the somatosensory cortex that receive inputs from the thalamocortical axons and could be mentioned in the discussion.

We believe that the images provided are a strong argument for TC arbors being shaped by RGC bouton distributions. We agree that reporting the correlation between dendrites and RGC boutons would be useful, but we found this correlation difficult to quantify. One of the challenges is that we would need to perform several-fold more reconstruction of dendrites and RGC boutons to have an unbiased mapping of both. Currently, most of the reconstructions stop when the dendrites assume a distal morphology and stop interacting with RGC boutons. Likewise, the EM of the RGC boutons are only those that innervate the reconstructed cells. We considered simply quantifying the asymmetry of the TC arbors relative to a symmetrical distribution and a random distribution, but we felt that quantification would be difficult to interpret without a similar analysis performed in the same region of dLGN on wild-type TCs.